

# Predicting suitable habitat of the Chinese monal (*Lophophorus lhuysii*) using ecological niche modeling in the Qionglai Mountains, China

Bin Wang[1,*], Yu Xu[2,3,*] and Jianghong Ran[1]

[1] Sichuan University, Key Laboratory of Bio-Resources and Eco-Environment of Ministry Education, College of Life Sciences, Chengdu, China
[2] Guizhou Normal University, College of Life Sciences, Guiyang, China
[3] Pingdingshan University, School of Resources and Environmental Sciences, Pingdingshan, China
[*] These authors contributed equally to this work.

## ABSTRACT

Understanding the distribution and the extent of suitable habitats is crucial for wildlife conservation and management. Knowledge is limited regarding the natural habitats of the Chinese monal (*Lophophorus lhuysii*), which is a vulnerable Galliform species endemic to the high-montane areas of southwest China and a good candidate for being an umbrella species in the Qionglai Mountains. Using ecological niche modeling, we predicted current potential suitable habitats for the Chinese monal in the Qionglai Mountains with 64 presence points collected between 2005 and 2015. Suitable habitats of the Chinese monal were associated with about 31 mm precipitation of the driest quarter, about 15 °C of maximum temperature of the warmest month, and far from the nearest human residential locations (>5,000 m). The predicted suitable habitats of the Chinese monal covered an area of 2,490 km$^2$, approximately 9.48% of the Qionglai Mountains, and was highly fragmented. 54.78% of the suitable habitats were under the protection of existing nature reserves and two conservation gaps were found. Based on these results, we provide four suggestions for the conservation management of the Chinese monal: (1) ad hoc surveys targeting potential suitable habitats to determine species occurrence, (2) more ecological studies regarding its dispersal capacity, (3) establishment of more corridors and green bridges across roads for facilitating species movement or dispersal, and (4) minimization of local disturbances.

Corresponding author
Jianghong Ran,
ranjianghong@scu.edu.cn

## INTRODUCTION

Understanding the distribution of suitable habitats and its influencing factors are crucial for wildlife conservation and management (*Austin, 2002*). Ecological niche models have developed as excellent tools for predicting habitat distribution of species that are difficult to investigate (*Peterson, Ball & Cohoon, 2002*; *Mota-Vargas et al., 2013*), because they can predict the distribution of species' habitats at a large spatial scale based on species presence data and environmental variables, without the need for extensive surveys and detailed

descriptions of physiological and behavioral characteristics (*Morrison, Marcot & Mannan, 2012*). In particular, MaxEnt, a presence-only modeling approach based on the maximum entropy principle (*Phillips, Anderson & Schapire, 2006*), outperforms other models in prediction accuracy (*Elith et al., 2006*; *Phillips, Anderson & Schapire, 2006*), transferability (*Tuanmu et al., 2011*), and performance with small sample sizes (*Pearson et al., 2007*; *Costa et al., 2010*). It offers great potential for addressing endangered and poorly known bird species with scarce occurrence data (e.g., *Botero-Delgadillo, Páez & Bayly, 2012*; *Marcondes et al., 2014*; *Tobón-Sampedro & Rojas-Soto, 2015*).

The Chinese monal (*Lophophorus lhuysii*) is the largest Galliform species (with a mean length of 76 cm and a mean weight of 3.18 kg) distributed in high-montane regions, mainly inhabiting subalpine scrubs, as well as subalpine and alpine meadows at an elevation of 3,000–4,900 m (*MacKinnon, Phillipps & He, 2000*; *Madge, McGowan & Kirwan, 2002*). This species is endemic to southwest China, and is found primarily in southeast Gansu, southeast Qinghai, western Sichuan, and northwest Yunnan (*Lei & Lu, 2006*; *Lu, 2015*). It has been listed on appendix I of CITES since 1975 (*CITES, 2016*) and classified as a vulnerable species on the IUCN red list since 1994 (*IUCN, 2015*). In 1989, the Chinese government started to legally protect the species as a first-class, nationally-protected wildlife species (*Ministry of Forestry of People's Republic of China, Ministry of Agriculture of People's Republic of China, 1989*). Recently, the Chinese monal was identified as one of the endemic bird species in China with the highest conservation values in terms of phylogenetic diversity (*Chen, 2013*). Its population size, however, remains small (10,000–25,000 individuals in total, *BirdLife International, 2015*) and is inferred to be in a state of continuous decline because of illegal hunting and on-going habitat degradation and fragmentation (*BirdLife International, 2015*). It is therefore necessary to establish effective conservation programs targeting the Chinese monal.

However, knowledge of the natural habitats of the Chinese monal is limited, as cryptic behavior and inaccessible habitats of the species make it difficult to survey the natural populations. Most of the previous studies were conducted in the late twentieth century and only provided simple descriptions of habitat use (*Ma, 1989*; *Zhang, 1995*), population density and structure (*He & Lu, 1985*; *Long et al., 1998*), breeding ecology (*He et al., 1986*), activity pattern (*Ma, 1988*), and feeding habits (*Lu et al., 1986*). There has not been a quantitative analysis of the natural habitats of the Chinese monal at a large spatial scale in any part of its geographical range and this limited knowledge has constrained conservation development for the Chinese monal.

The Qionglai Mountains are a part of the biodiversity hotspot of the Mountains of Southwest China (*Mittermeier et al., 2011*) that supports many endangered and endemic wildlife species. As the geographical center of the distribution of the Chinese monal (*Lei & Lu, 2006*; *Lu, 2015*; Fig. 1), this region is critical for the conservation of the species. In this study, we used the ecological niche modeling approach, with species records and predictors selected from a large set of environmental variables, to predict the current potential suitable habitats for the Chinese monal in the Qionglai Mountains. Our objectives were to (1) delineate the distribution of suitable habitats of the Chinese monal, (2) identify

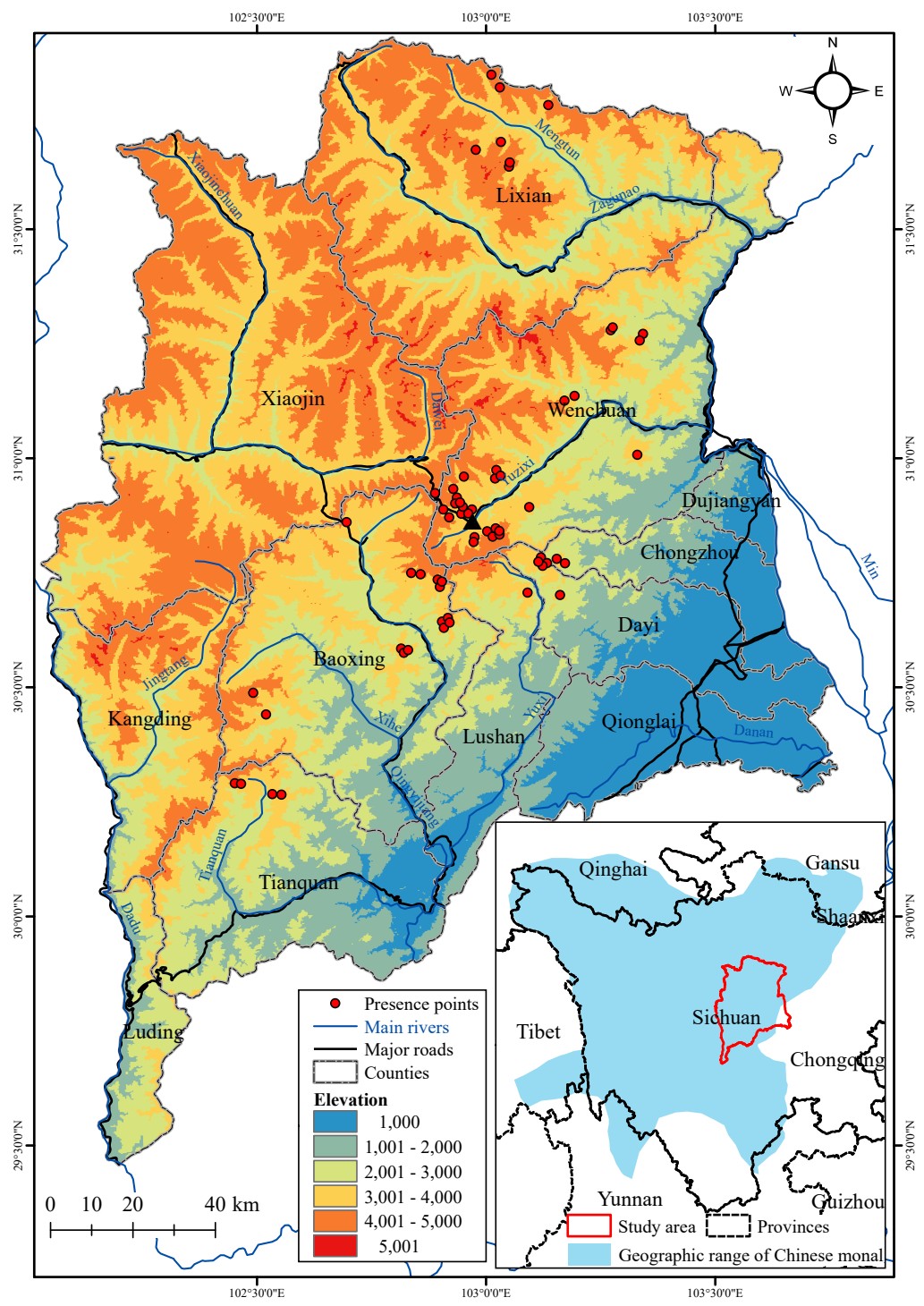

**Figure 1** **Topographic map of the Qionglai Mountains, showing the location of the 64 presence points of the Chinese monal used in modeling.** The black triangle shows the location of Huayan tunnel at which we observed monals crossing the road daily (detailed in 'Discussion'). The geographic range of Chinese monal was delineated based on *Lu (2015)*.

critical environmental factors influencing the species' habitat suitability, (3) compare the prediction with existing nature reserve network and provide conservation suggestions based on the results.

## MATERIALS & METHODS

### Study area

The Qionglai Mountains are the easternmost portion of the Hengduan Mountains, extending approximately 250 km from north to south at the center of Sichuan Province, China (E: 102°01′–103°46′, N: 29°27′–31°55′; Fig. 1). The area covers 26,258 km$^2$ and incorporates 12 counties, including Lixian, Xiaojin, Baoxing, Dayi, Lushan, Qionglai, and Tianquan; the areas west of the Min River in Wenchuan, Dujiangyan and Chongzhou counties are also included in the Qionglai Mountains (the east side of the River belongs to the Minshan Mountains); the areas east of the Dadu River in Luding and Kangding counties are also included in the Qionglai Mountains (the west sides of the River belong to the Daxueshan Mountains). Elevation ranges from 450 m in the Sichuan Basin to 6,250 m at the Siguniang Mountain peak. The subtropical monsoon climate predominates the region. There are diverse ecosystems and vegetation types, with many rare and endangered wildlife species, such as the giant panda (*Ailuropoda melanoleuca*), golden snub-nosed monkey (*Rhinopithecus roxellanae*), snow leopard (*Panthera unica*), takin (*Budorcas taxicolor*), and dove tree (*Davidia involucrate*) (*Hu, 2001*).

### Presence data

In total, we gathered 103 current (2005–2015) presence records of the Chinese monal (Table S1), including sightings, callings, feces, and feathers. The majority (86.4%) of the records was from field surveys in recent years, including the authors' filed survey records, sympatric animal database of the fourth national survey on giant panda, and biodiversity monitoring data from nature reserves; the remaining data was from published literatures and network databases (for details, see Table 1). To insure spatial accuracy, we only kept presence points with accuracies to three decimal places of the coordinates. To avoid overfitting the model, we generated a 1 km$^2$ buffer around each presence point and randomly selected one if the buffers overlap, as this area approximates the minimum home range size maintained by the Chinese monal (*He & Lu, 1985*). Finally, 64 remained presence points with geographic coordinates were used to build the ecological niche model (Fig. 1).

### Environmental data

A total of 42 environmental variables (Table S2) summarizing five groups (bioclimatic data, vegetation, phenological metrics, topographical attributes, and human impacts) potentially related to habitat suitability of the Chinese monal were selected as candidate variables. We included 19 bioclimatic variables from WorldClim 1.4 (http://www.worldclim.org/), which were interpolated based on a large number of weather stations all over the world, integrating the effects of latitude, longitude, and elevation (*Hijmans et al., 2005*). Bioclimatic data were frequently used in habitat modeling due to direct effects on species distribution (*Guisan*

**Table 1** Sources of Chinese monal presence data in the Qionglai Mountains used in the suitable habitat modeling.

| Source | Elevational range and area of survey site | Survey technique and effort | Survey time | Used/total presence points |
|---|---|---|---|---|
| Global Biodiversity Information Facility (http://www.gbif.org/) | Network database | Bird-watching records | 2005–2015 | 2/7 |
| Shen, Li & Xiang (2010) | 3,300–4,170 m; 35 km² | Published literature | Apr–Nov, 2007–2010 | 7/7 |
| Comprehensive scientific survey on Anzihe Nature Reserve | 1,638–3,868 m; 110 km² | Line transect; 29 3–4.9 km transects | Apr–Oct, 2010 | 3/3 |
| Comprehensive scientific survey on Fengtongzhai National Nature Reserve | 1,000–4,896 m; 403 km² | Line transect; 38 2–5 km transects | May–Oct, 2010–2011 | 15/28 |
| Sympatric animal database of the fourth national survey on giant panda | 1,000–4,400 m; 8,740 km² | Grid square; 4,370 2 km² squares | Mar–Dec, 2012–2013 | 22/24 |
| Infrared-triggered camera monitoring in Heishuihe Nature Reserve | 1,520–4,234 m; 234 km² | Infrared-triggered camera; 70 cameras | All year round, 2013–2014 | 5/15 |
| Infrared-triggered camera monitoring in Labahe Nature Reserve | 1,500–4,500 m; 170 km² | Infrared-triggered camera; 30 cameras | All year round, 2014–2015 | 4/5 |
| Montane bird survey in Wolong National Nature Reserve | 3,600–4,400 m; 17 km² | Line transect; 7 2–6 km transects | Jun–Jul, 2015 | 6/14 |

& Zimmermann, 2000). Vegetation and phenology-related variables were derived from moderate resolution imaging spectroradiometer (MODIS) remotely sensed data from LP DAAC (https://lpdaac.usgs.gov/). The International Geosphere-Biosphere Programme (IGBP) classification of land cover within MCD12Q1 2014 product was used as a categorical variable of vegetation type (Loveland & Belward, 1997). This classification was derived from yearly MODIS data and 1860 training sites observations distributed across the Earth's land areas (Friedl et al., 2010). We downloaded a time series of MOD13Q1 16-day enhanced vegetation index (EVI) product over a three year period from 2013 to 2015 (23 layers per year). To reduce the potential noise caused by cloud remnants, we reconstructed a clean and smooth EVI time series employing an adaptive Savitzky-Golay filter and then generated 15 phenological metrics using TIMESAT 3.2 (Jönsson & Eklundh, 2002; Jönsson & Eklundh, 2004). These EVI derived metrics strongly correlate with vegetation primary productivity and its seasonality (Jönsson & Eklundh, 2002; Rahman et al., 2005; Alcaraz-Segura et al., 2013), and could improve the performance of species habitat modeling (Requena-Mullor et al., 2014). The topographical attributes included elevation, slope, aspect, and distance to rivers. Slope and aspect were extracted from an ASTER GDEM V2 30 m resolution digital elevation model (DEM; http://www.gscloud.cn/), using the Surface Analyst Tool of ArcGIS 10.2 (ESRI, Redlands, CA, USA). Aspect was recalculated as the absolute value of actual degree minus 180°, representing how close the slope was to the adret facing the sun (Kalkhan, 2011). We produced a layer of distance to the nearest perennial river using the Euclidean Distance Tool of ArcGIS. Variables regarding human impacts included Euclidean distance to residential locations (villages and rural settlements) and to roads, and human influence index (HII). HII layer was downloaded from Last of the Wild (Data Version 2,

*2005)* (http://sedac.ciesin.columbia.edu/), representing anthropogenic impacts spanning 1995–2004 that were calculated by integrating the data including human population pressure (population density), human land use (built-up areas, nighttime lights, land use, and land cover), and human accessibility (coastlines, roads, railroads, and navigable rivers). The basic vector layers of rivers, roads, and residential locations were provided by National Geomatics Center of China (NGCC). Each variable was projected to the UTM zone 48N coordinate system, and resampled to the same pixel size as the EVI layers (about 250 m) using bilinear interpolation, except for the categorical land cover that was resampled using nearest neighbor assignment.

## Model procedure

We used MaxEnt to generate the habitat suitability model of the Chinese monal (MaxEnt 3.3.3k, http://biodiversityinformatics.amnh.org/open_source/maxent/). Ecological niche models with inappropriately complex variables might be oversized, overfitted or redundant (*Parolo, Rossi & Ferrarini, 2008*; *Swanepoel et al., 2013*). To increase abilities in building high accuracy predictions and in identifying the critical predictors constraining the species' distribution, we implemented an optimized selection of 42 environmental variables based on sample-size-corrected Akaike information criteria (AICc) (*Akaike, 1974*; *Warren & Seifert, 2011*; *Warren et al., 2014*). Firstly, we built a MaxEnt model with the full set of 42 variables and removed variables with contribution <1% or had Pearson's correlation coefficients >|0.7| with the highest contributed variable; then the retained variables were used to build a new model, and the variables with low contribution (<1%) or high correlation coefficients (absolute values > 0.7) with the second highest contributed variable were removed again; finally, a set of models with different set of variables were produced after repeating this process, and AICc value was calculated for each model based on codes proposed by *Warren, Glor & Turelli (2010)*. The model with the lowest AICc was considered to have the most appropriate complexity (*Warren & Seifert, 2011*; *Jueterbock et al., 2016*), thus the variables included in this model were selected to build the final model for Chinese monal habitat. The optimized variable selection was processed in R 3.2.2 (*R Development Core Team, 2015*) with package "MaxentVariableSelection" (*Jueterbock, 2015*; *Jueterbock et al., 2016*).

The regularization multiplier, maximum number of background points, maximum iterations, and convergence threshold were set as default values, since these settings have been found to achieve good performances (*Phillips & Dudík, 2008*). To produce stable results, we ran 20 replicate bootstrap procedures for the final model. Each replicate used a randomly selected dataset of 75% training data and 25% test data. The built-in functions: contribution of variables, response curves, and jackknife tests were used to analyze the relative importance of each variable in modeling, and their relation with the habitat suitability. We used an average output grid of 20 replicates as the final model prediction, with a logistic habitat suitability index ranging from the lowest "0" to the highest "1."

Model performance was evaluated using three different measures, including area under the receiver operating characteristic curve (AUC), Cohen's maximized Kappa, and the true skill statistic (TSS). All three measures are calculated based on specificity and sensitivity

of the predictive model. Specificity and sensitivity represent the success rate for classifying absences and presences, respectively. AUC is a threshold-independent evaluation measure obtained by plotting sensitivity against 1-specificity (*Fielding & Bell, 1997*). We used the built-in function of MaxEnt program to produce the mean AUC values of the 20 replicates. The model accuracy can be judged as excellent if AUC > 0.9, good if 0.9 > AUC > 0.8, fair if 0.8 > AUC > 0.7, poor if 0.7 > AUC > 0.6, and failed if 0.6 > AUC > 0.5 (*Swets, 1988*).

Kappa and TSS are threshold-dependent indices that measure the agreement between predictions and known occurrences (presences and absences) at different binary thresholds. Kappa index is obtained by plotting sensitivity and specificity against different thresholds (*Cohen, 1960*), while TSS equals sensitivity + specificity − 1 (*Allouche, Tsoar & Kadmon, 2006*). Another difference is that Kappa index responds to species prevalence whereas TSS does not (*Allouche, Tsoar & Kadmon, 2006*). In this study, we used maximized values of Kappa and TSS at their own optimal thresholds to evaluate model performance. The standards for judging model performance are: excellent if Kappa > 0.75, good if 0.75 > Kappa > 0.4, and poor if Kappa < 0.4 (*Araújo et al., 2005*); while good to excellent if TSS > 0.8, useful if 0.8 > TSS > 0.5, and poor if 0.5 > TSS > 0.2 (*Coetzee et al., 2009*). Since these two indices require the use of absence data, 200 additional pseudo-absence points were generated within the study region. These pseudo-absence points were randomly located outside the 1 km buffers of observed presence points for Chinese monals. The threshold-dependent statistics were analyzed in R 3.2.2 (*R Development Core Team, 2015*) with package "PresenceAbsence" (*Freeman & Moisen, 2008*).

## Habitat analysis

For further analysis, we applied a threshold that maximizes the TSS for transforming the model prediction with a continuous habitat suitability index to a binary suitable/unsuitable map. Maximum TSS is a promising threshold criterion when only species presence data are available, outperforming many other criterions in most cases (*Liu et al., 2005*; *Jiménez-Valverde & Lobo, 2007*; *Liu, White & Newell, 2013*). We estimated total suitable habitat area of the Chinese monal in the Qionglai Mountains and areas harboured in counties and nature reserves, respectively.

## RESULTS

Optimized variable selection showed that the model with the lowest AICc was built with: maximum temperature of the warmest month (bioclim5), precipitation of the driest quarter (bioclim17), annual maximum of EVI (evi maximum), base level values of EVI (evi base level), slope, and distances to residential locations (d_resident) and to roads (d_road) (Tables S3 and S4). The seven predictors were therefore selected for the final MaxEnt modeling. We obtained average values of model evaluation indices after 20 replicates: training AUC = 0.966, test AUC = 0.953, maximum Kappa = 0.813, and maximum TSS = 0.882. High scores of model evaluation indices, both the threshold-independent and threshold-dependent, indicated that the habitat suitability model produced by MaxEnt performed excellently (*Manel, Williams & Ormerod, 2001*; *Araújo et al., 2005*).

**Table 2** Relative importance of environmental variables in the habitat suitability model of the Chinese monal.

| Variables | Percent contribution[a] | Jackknife of AUC[b] |
|---|---|---|
| bioclim17 | 36.2 | 0.856 |
| bioclim5 | 31.0 | 0.840 |
| d_resident | 10.5 | 0.771 |
| evi base level | 8.8 | 0.689 |
| evi maximum | 6.0 | 0.683 |
| slope | 4.8 | 0.631 |
| d_road | 2.7 | 0.642 |

Notes.

[a]The relative contribution of each variable to predictive model, shown as mean value of 20 replicates.

[b]Jackknife test of variable importance, expressed as AUC (area under the receiver operating characteristic curve) for models using each variable alone. A higher gain indicates a variable with more information for modelling when used in isolation, shown as mean value of 20 replicates.

bioclim17, precipitation of the driest quarter; bioclim5, maximum temperature of the warmest month; d_resident, distance to residential location; evi base value, annual base level value of EVI; evi maximum, annual maximum EVI; d_road, distance to roads.

## Relevant variables

Evaluation of percent contribution of each variable to the model illustrated that three predictors had over 10% relative contribution to the habitat suitability model of the Chinese monal, contributing 78% collectively (Table 2). Precipitation of the driest quarter (bioclim17) made the largest contribution, followed by maximum temperature of the warmest month (bioclim5) and distance to the nearest residential locations. Similarly, these three predictors received the highest AUC values when used in isolation in the jackknife test (Table 2), indicating they are better at discriminating suitable from non-suitable habitat as compared with the other variables. The response of logistic probability of monal occurrence on the two critical climatic predictors were both unimodal, that the probabilities were at the peak at 31 mm for precipitation of the driest quarter and at 15 °C for maximum temperature of the warmest month (Figs. 2A and 2B). The response curve also showed that monals were more likely to occur at sites further away (>5,000 m) from the nearest human residential locations (Fig. 2C).

## Habitat status

We created a binary suitable/unsuitable habitat map after applying the maximum TSS threshold (where habitat suitable index = 0.208). The suitable habitats for the Chinese monal covered an area of 2,490 km$^2$, corresponding to 9.48% of the entire Qionglai Mountains region. The majority of suitable habitats for the Chinese monal distributed in the central mountain regions, mainly stretching along the boundaries of Lixian-Wenchuan, Wenchuan-Baoxing, Xiaojin-Baoxing, and Baoxing-Tianquan (Fig. 3, Table 3). Ten existing nature reserves are located in the Qionglai Mountains, covering over a quarter of the entire region. Over 50% of the suitable monal habitats were situated within nature reserves. Wolong Nature Reserve provided the largest area of suitable habitats for the Chinese monal, while Heishuihe, Labahe, and Fengtongzhai Nature Reserve also had high proportions of suitable habitat (Table 4). Two general regions with large areas of potential suitable habitat
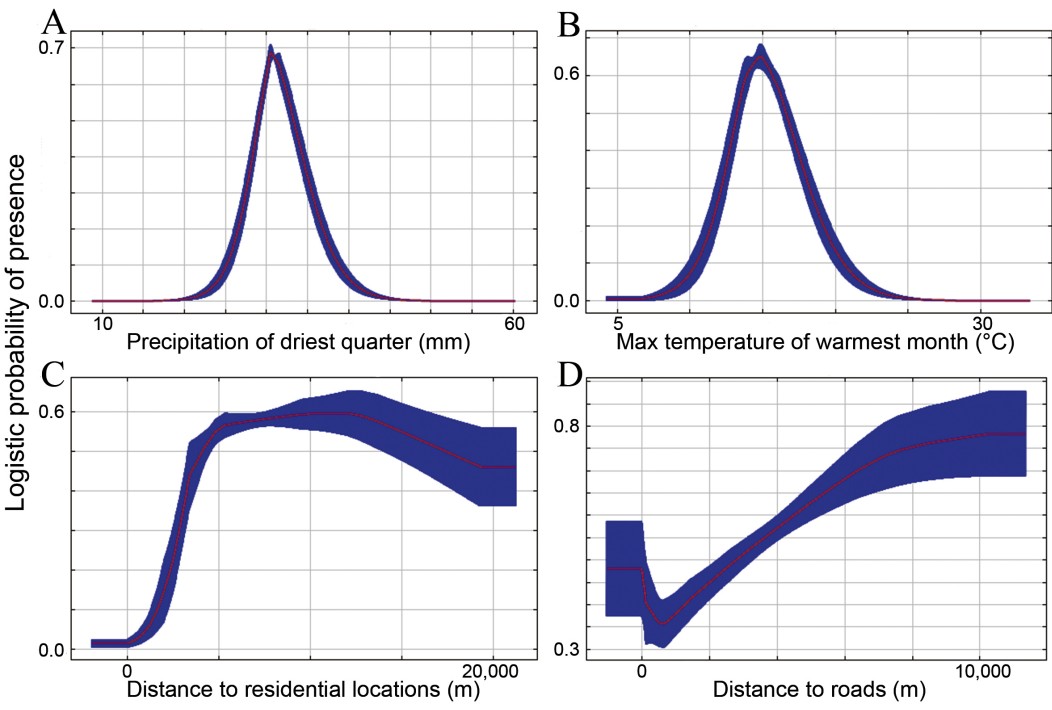

**Figure 2   Response curves of habitat suitability for the Chinese monal (vertical axis) to the precipitation of the driest quarter (A), maximum temperature of the warmest month (B), distance to residential locations (C), and distance to roads (D).** The red lines illustrate the mean responses of 20 replicates and the blue shades showed the ± standard deviation.

were outside of any nature reserves: one in the northern region of Lixian County, and the other in northwest Baoxing County (Fig. 3).

## DISCUSSION

Our study represents the first attempt at predicting the suitable habitat of the Chinese monal. Variable analysis revealed that bioclimatic variables were the most influential predictors on the habitat suitability of the Chinese monal. Higher habitat suitability was constrained in narrow ranges of both precipitation of the driest quarter and maximum temperature of the warmest month, suggesting that Chinese monal was highly sensitive to climate under extreme periods. The monals appeared to prefer habitats far away from residential locations, suggesting human disturbance as a crucial pressure for the species. Although roads represent remarkable sources of disturbance to wildlife that could lead to road-kill (*Mumme et al., 2000*) and barrier effect (*Shepard et al., 2008*), the impact of roads was relatively weak in our niche model (Table 2). It is likely that a greater part of occurrence records were sampled in more easily accessible sites in proximity to roads (Fig. 1), and such a sampling bias had decreased the importance of roads disturbance in the model. Despite the sampling bias towards roads, we found an relation that habitat suitability increased with increasing distance from roads in the niche model predictions (Fig. 2D). As an explanation, we suggest that our predictive habitat suitability was more sensitive
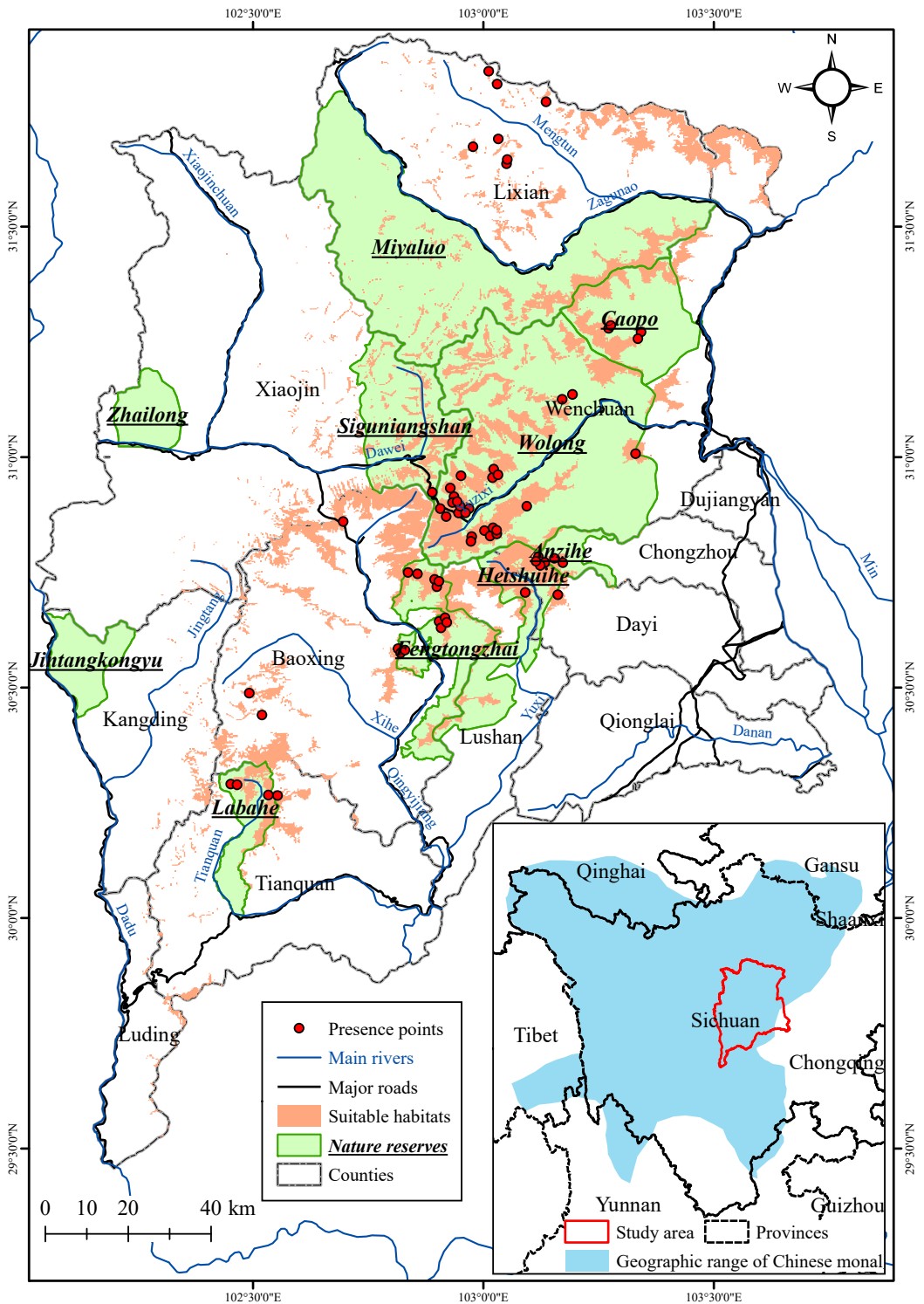

**Figure 3 Predicted suitable habitat for the Chinese monal and existing nature reserves in the Qionglai Mountains.** The geographic range of Chinese monal was delineated based on *Lu (2015)*.
**Table 3 Estimates of suitable habitat areas of the Chinese monal distributed in counties within the Qionglai Mountains.**

| County | County area (km²) | Suitable habitat area (km²) | Proportion of suitable habitat (%) |
|---|---|---|---|
| Wenchuan[a] | 3,589 | 852 | 23.74 |
| Baoxing | 3,124 | 544 | 17.41 |
| Lixian | 4,325 | 401 | 9.27 |
| Xiaojin | 5,568 | 235 | 4.22 |
| Lushan | 1,259 | 159 | 12.60 |
| Tianquan | 2,390 | 150 | 6.28 |
| Dayi | 1,207 | 80 | 6.63 |
| Kangding[b] | 1,666 | 37 | 2.22 |
| Chongzhou[a] | 803 | 19 | 2.37 |
| Luding[b] | 647 | 13 | 2.00 |
| Qionglai | 1,377 | 0 | 0 |
| Dujiangyan[a] | 303 | 0 | 0 |
| Total | 26,258 | 2,490 | 9.48 |

**Notes.**

[a] Wenchuan, Dujiangyan and Chongzhou described here are only their west parts of the Min River.
[b] Kangding and Luding described here are only their east parts of the Dadu River.

to bioclimatic conditions than to human disturbance; sites with bioclimatic conditions preferred by the Chinese monal might be infrequently crossed by roads.

The predicted suitable habitats for the Chinese monal were highly fragmented. Specialized niche requirements (i.e., narrow climate preferences) constrained their suitable habitats to subalpine and alpine regions. High mountain ridges and deep valleys throughout the Qionglai Mountains separated the suitable haibtats into small and isolated patches, especially for those distributed in the north of Lixian County, south of Baoxing County, and the boundary area of Lixian-Xiaojin-Wenchuan County (Fig. 3). As a strong disturbance, human residential areas further restricted and separated the suitable habitats. The fragmentation might have negative effects on the Chinese monal, which has low dispersal ability (*With & Crist, 1995*), reducing the probability of genetic exchange (*Höglund et al., 2011*) and colonizing suitable empty patches (*Stamps, Buechner & Krishnan, 1987*) by increasing the resistance to individual movement between isolated habitat patches.

Many subalpine meadows within the Qionglai Mountains are summer pasture for the grazing yaks. Grazing could lead to direct disturbance, accelerating habitat degradation and fragmentation, and increasing the probability of poaching (*Lu et al., 1986*; *Ma, 1988*; *Long et al., 1998*). Additionally, gathering of herbs for use in traditional Chinese medicine in spring and summer, especially the collection of *Fritillaria* spp., an important food source of the Chinese monal, could result in a reduction of food abundance and thus a indirect disturbance to the Chinese monal (*Fuller & Garson, 2000*; *Lu, 2015*). Our study, however, failed to incorporate these remarkable local threats into modeling due to the lack of data. If grazing and herb gathering are considered, the actual suitable monal habitat might be even smaller, more fragmented, and more variable than our prediction.
**Table 4   Estimates of suitable habitat areas of the Chinese monal in nature reserves.**

| Nature reserve | Reserve area (km²) | Suitable habitat area (km²) | Proportion of suitable habitat (%) |
|---|---|---|---|
| Wolong | 2,124 | 585 | 27.54 |
| Miyaluo | 1,951 | 181 | 9.28 |
| Heishuihe | 325 | 146 | 44.92 |
| Siguniangshan | 583 | 131 | 22.47 |
| Caopo | 517 | 128 | 24.76 |
| Fengtongzhai | 403 | 111 | 27.54 |
| Labahe | 239 | 69 | 28.87 |
| Anzihe | 110 | 18 | 16.36 |
| Jintangkongyu | 242 | 0 | 0 |
| Zhailong | 204 | 0 | 0 |
| Total | 6,698 | 1,364 | 20.36 |

Our results raise concerns for the status of the Chinese monal in the Qionglai Mountains, and the species may be at greater risk than has previously been considered. As a typical high-montane species with large home range, the Chinese monal has the potential to serve as an umbrella or flagship species for high-montane ecosystems. Conservation of these ecosystems will likely contribute to the maintenance of regional biodiversity, at least for the montane Galliformes (*Roberge & Angelstam, 2004*; *Rowland et al., 2006*; *McGowan, Zhang & Zhang, 2009*). Based on our results, we have several conservation suggestions targeting the Chinese monal. Our prediction could guide future field surveys for locating new populations in the areas that were predicted to be suitable habitat but lack investigation (*Raxworthy et al., 2003*; *Menon et al., 2010*). The large and well-connected habitat patches distributed in the boundary area of Wenchuan, Baoxing, Lushan, and Dayi County appear to provide ideal and important habitats for the Chinese monal and should be treated as a priority area for conservation. However, a large portion stretching along the northern boundary of Baoxing County is unprotected by any nature reserves and lacks systematic survey (Fig. 3). Similarly, large ribbon-like patches of suitable habitats in northeast of Lixian County and north of Wenchuan County was also lacking protection and survey (Fig. 3). Our first suggestion, therefore, is to conduct ad hoc surveys for determining the actual occurrence of Chinese monal in these two areas. If there are indeed some populations in these two regions, we suggest that new nature reserves should be designated in order to fill these two obvious conservation gaps. The second suggestion is to study the dispersal capacity of the Chinese monal for further assessing habitat quality and fragmentation. Such ecological knowledge could be helpful for determining the actual distribution of animals in relation to the distribution of suitable environmental habitats (*Pulliam, 2000*; *Lu et al., 2012*). For instance, it will be useful to determine whether the Zagunao river valley is a geographic barrier that Chinese monals cannot cross through, and whether gene flow can occur between the monal populations isolated in the south and north of Baoxing County (Fig. 3).

Although human disturbance appeared to be a crucial pressure for Chinese monals (*He et al., 1986*), we found that Chinese monals did not respond avoiding roads. For instance, during our survey at Wenchuan in 2015, we observed that two Chinese monal pairs, almost at a daily bases, crossed provincial road 303 through the top of the Huayan tunnel (E102°58′, N30°51′; Fig. 1), a well-known location for shooting Chinese monal among wildlife photographers. Huayan tunnel has a length of 570 m, and the large area of natural shrub and meadow preserved on its top serves as a corridor for the monals. However, this does not mean that roads are not a significant threat for the species, as road-kill of Galliformes are not uncommon (*Clevenger, Chruszcz & Gunson, 2003*; *D'Amico et al., 2015*). Our third suggestion, therefore, is to build more corridors and green bridges across roads to facilitate road-crossing or dispersal by Chinese monals.

For the purpose of 'game food', Chinese monals had been often poached by local people before 1980s, which was considered as the main cause of a substantial decline of the species (*He et al., 1986*; *Lu et al., 1986*; *Long et al., 1998*; *Lei & Lu, 2006*). Unfortunately, we found self-made traps for capturing monals and other Galliformes during our surveys, both inside and outside nature reserves, suggesting that poaching is still continuing nowadays. Even though we found little evidence of human impact on monal habitat distribution, our field observation suggested that illegal hunting potentially threaten survival of the Chinese monals and their sympatric Galliform species. Moreover, other short-term and seasonal local disturbance such as yak grazing and herb collection could increase the probability of poaching (*Lu et al., 1986*; *Ma, 1988*; *Long et al., 1998*). Therefore, our fourth suggestion is to limit local disturbances, such as poaching, yak grazing, and herb collection, for maintaining existing populations and habitats of the Chinese monal as well as other montane Galliformes.

## CONCLUSION

This study used the ecological niche modeling approach to predict current suitable habitat of the Chinese monal in Qionglai Mountains. Their suitable habitat was associated with about 31 mm precipitation of the driest quarter, about 15 °C of maximum temperature of the warmest month, and far from the nearest human residential locations (>5,000 m). The predicted suitable habitats of the Chinese monal was highly fragmented covering an area of 2,490 km$^2$. A total of 54.78% of suitable habitat was under the protection of existing nature reserve network, but there were obvious conservation gaps as two regions with large area of well-connected suitable habitats were out of any nature reserve. Finally, this study provide conservation management suggestions in terms of ad hoc surveys targeting potential suitable habitats to determine occurrence of Chinese monals, more ecological studies regarding its dispersal capacity, establishment of more corridors and green bridges across roads in suitable habitats and limitation of local disturbances such as poaching, yak grazing, and herb collection.

## ACKNOWLEDGEMENTS

We thank Heishuihe Nature Reserve and Labahe Nature Reserve for providing monitoring data, and Fengtongzhai National Nature Reserve, Wolong National Nature Reserve, Anzihe Nature Reserve for the supports provided during the fieldworks. We are grateful to Qiang Dai and Tingting Yan for their recommendations on methodology, and Liang Dou, Nan Yang, Xiang Yu, Bo Zhang for their assistance in the fieldworks. Zhengyang Wang from Harvard University helped to correct the English writing of the manuscript.

### Funding

Funding for this research was provided by the National Natural Science Foundation of China (No. 31301896); the New Teacher Fund for Doctor Station of Ministry of Education of China (No. 20120181120087). There was no additional external funding received for this study. The funders had no role in study design, data collection and analysis, decision to publish, or preparation of the manuscript.

### Grant Disclosures

The following grant information was disclosed by the authors:
National Natural Science Foundation of China: 31301896.
New Teacher Fund for Doctor Station of Ministry of Education of China: 20120181120087.

### Competing Interests

The authors declare there are no competing interests.

### Author Contributions

- Bin Wang performed the experiments, analyzed the data, contributed reagents/materials/analysis tools, wrote the paper, prepared figures and/or tables.
- Yu Xu conceived and designed the experiments, performed the experiments, reviewed drafts of the paper.
- Jianghong Ran conceived and designed the experiments.

### Data Availability

The raw data of this research is uploaded as Table S1.

### Supplemental Information

Supplemental information for this article can be found online at http://dx.doi.org/10.7717/peerj.3477#supplemental-information.

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
