# Peer review of "Predicting suitable habitat of the Chinese monal (Lophophorus lhuysii) using ecological niche modeling in the Qionglai Mountains, China"

_PeerJ, doi:10.7717/peerj.3477_

## Round 0.1 · original submission · Major Revisions

As usual for manuscripts submitted for publication as an Original Paper, your paper has been evaluated by two peer reviewers. Concerning this submission, however, they are both quite critical, express major concerns with your ms and unambiguously advise against publication of your work as is. Most importantly, their criticisms touch not only shortcomings in the presentation of your work (starting with the need of a thorough language makeover) but they also provide some general comments on more severe issues, the proper addressing of which would call for a very substantial revision of your ms. English language is not perfect. At several places, articles are missing or other smaller grammatical mistakes make the text more difficult to understand. You also have to clarify why the selected variables are the ones that best represent habitat suitability (see Rev2 ), and to clarify why your analysis is not an over –interpretation as is suggested by Rev 2.

When revising your ms, I sincerely ask you to adequately address EACH of the referees' comments, as well as EACH of my general remarks on format issues, by either incorporating the suggestions in the revision, if possible, or providing brief but convincing rebuttals in case you do not agree with them. The referees' comments can be found at the end of this email.

Reviewer 1 ·

Basic reporting

Overall, this was a very interesting manuscript written in a professional way with very nice figures supporting data. However, English language could be improved. Specifically, I noticed an overuse of articles. Even though this does not mean that comprehension is difficult I consider it could be written in a better way. Some examples where the language could be improved include lines 34, 36-38, 57-58, 107, 108, 129, 209, 228, 267.
(Please check my specific annotated suggestions on the text)

Experimental design

In general, experimental design is clear. I commend the authors for their extensive data set on presence points, compiled over 10 years of fieldwork and published literature. Regarding, this last source of data I wonder whether results vary if authors exclude from the analysis those 12 records obtained from network database and published literature (since those presence points may be less reliable than your own fieldwork observations). In lines 95-97 I would also suggest to better explain how some points were discarded from analysis.

Validity of the findings

Very interesting conservation recommendations are derived from results of this study. I would only recommend to be careful with some statements declared in the discussion section (lines 236, 281-281) because some are not directly supported by results.

Additional comments

The manuscript by Wang et al. deals with a chinese galliform bird (Chinese monal) threatened by illegal trapping and habitat loss. The ecological niche modeling aims to distinguish potential suitable habitats for this species and provide suggestions for conservation management.

Even though it is considered VU by the IUCN, the species has high conservation value in terms of phylogenetic diversity, meaning that the species contributes in genotypic, phenotypic and/or functional diversity to the clade. The study presents a possible valuable contribution towards the conservation of this impressive species. In particular, I see the quantification of possible suitable habitat of high value for conservation action plans. To this end, authors though should extent their recommendations to describe in a more specific way actions for the conservation of the Chinese monal based on the results of their study.

Apart from its general structure, the manuscript would benefit from streamlining and an extensive language editing. While most of the English is ok, the writing definitely should be improved as the text is rather hard to read at some parts. Consultation of a native colleague with also good writing skills is recommended.

Overall, this was a very interesting manuscript. My specific comments are annotated on the text, which mainly include a few suggestions to improve flow and readability, and some considerations for the Discussion.

Annotated reviews are not available for download in order to protect the identity of reviewers who chose to remain anonymous.

Reviewer 2 ·

Basic reporting

The raw data are supplied but Supplementary Table 1 should state which
of the 98 occurrence records were removed to avoid overfitting and
which are the 70 records that were used for niche modeling.

English language is not perfect. At several places, articles are
missing or other smaller grammatical mistakes make the text more
difficult to understand. For example, line 129 change 'integrated' to
'integrating'. Line 134: remove 'method'. Line 150: change 'max' to
'maximum'. Line 226: 'represent' should be changed to
'represents'. Line 228: Change 'Suitability were constrained' to
'suitability was constrained'. Line 266-67 change 'a large
portion'...'are unprotected' to ...'is unprotected'.

Experimental design

The authors identify very well the knowledge gap that this study aims
to fill.

The identification of the most important environmental factors is
unclear. Variables were removed with contribution <0.1% or correlation
>0.7 with the help of an R package but whether removal of the
variables increased or decreased model performance is not
stated. Model performance is only evaluated for the minimum set of
independent, variables with a model contribution >1%. This is the major
weakness of the manuscript. To me it is not clear why the selected
variables are those that best predict habitat suitability.

Validity of the findings

A conclusion section is missing. I would recommend the authors to
include one.

In my opinion, predictions for suitable habitat from niche model
predictions are not sufficient to alter the location of conservation
areas. The suitability of habitats that appear suitable but are not yet
under protection should first be confirmed by field studies.

Additional comments

The study presents the results of primary scientific research. The
authors develop an ecological niche model for the Chinese monal in
order to identify whether protected areas in the Qionglai Mountains
overlap with regions of highest habitat sutiability. My major two
criticisms are: 1) It remains unclear to me why the selected variables
are the ones that best represent habitat suitability (see below). 2)
The suggestion to shift protection areas based on a single niche
modeling study is for me an over-interpretation of the data. That the
predicted habitat is indeed suitable should be first confirmed by
targeted field observations.

On the map (Figure 1) it appears that most sampling locations were
collected in proximity to roads. I would expect that this would create
a sampling bias towards more easily accessible sites (e.g. roads).
The authors should discuss why the inverse relation (increasing
habitat suitability with increasing distance from roads) was found in
the niche model predictions.

Why are only 4 of the 8 important variables shown in Fig. 2? Three of
them had over 10% relative contribution to the habitat suitability of
the Chinese monal. Why is the fourth one shown but not the other six?

The text describing the study area is difficult to follow without
figure. I would recommend to refer to Fig. 1 early in this
paragraph (for example line 78).


Line 190: ''unrelated' should be changed to 'not significantly
correlated' and 'relatively important' should be also specified in
more detail (importance >1%).

Line 204-205: 'Three predictors'...'appearing to have more useful
information by themselves' is very vague. Should be better changed to
something like: 'appear to better discriminate suitable from
non-suitable habitat as compared with the other variables'.

Line 218: Start sentence with the word 'Ten' instead of the number
'10'.

Line 274-275: It is unclear to me what 'can communicate by dispersing'
means. I guess it doesn't refer to vocal or visual communication but
to genetic exchange?

---

## Round 0.2 · accepted · Accept

It's my pleasure to inform you that, after the peer review, your paper, Predicting suitable habitat of the Chinese monal (Lophophorus lhuysii) using ecological niche modeling in the Qionglai Mountains, China; has been Accepted.